# Study on Characteristics of Pipeline Hydraulic Transportation of Coarse Particles Based on LBM-DEM Method

**Yuxi Wu [1,2], Weijing Niu [3], Tingting Zhao [1,2,\*] and Zhiqiang Li [1,2]**

1   College of Mechanical and Vehicle Engineering, Taiyuan University of Technology, Taiyuan 030024, China; wuyuxi0034@link.tyut.edu.cn (Y.W.); lizhiqiang@tyut.edu.cn (Z.L.)
2   Shanxi Key Laboratory of Material Strength & Structural Impact, Taiyuan 030024, China
3   Shanxi Polytechnic College, Taiyuan 030006, China; yanran001@126.com
\*   Correspondence: zhaotingting@tyut.edu.cn; Tel.: +86-176-3511-0963

**Abstract:** Pipeline hydraulic transportation is an important method for transporting solid materials. An LBM–DEM-based simulation method is established in the Visual Studio platform using the Fortran language, which can model the hydraulic transportation process of coarse particles in a pipeline. To verify the feasibility of this numerical simulation method, we also conduct pipeline hydraulic transportation experiments and find that the simulation results are consistent with the experimental results. This method is used to investigate the motion characteristics of coarse particles in horizontal pipelines under different conditions because of its advantages of clear physical meaning, ability to deal with complex boundary conditions, and suitability for parallel computing. The results show that the bedload motion is the main motion mode of coarse particles in horizontal pipelines. Changes in the pipeline flow velocity and particle diameter can cause some particles to saltate, and the critical pipeline flow velocity of saltation particles has a linear relationship with the particle diameter. During the process of coarse particles changing from the static state to stable state, the velocity curve of coarse particles gradually changes from "C" type to "S" type with the increase of particle diameter. Moreover, there is a linear relationship between the pipeline flow velocity and the stable velocity of coarse particles, while the particle diameter has no significant influence on the stable velocity. This study provides guidance for the development of pipeline hydraulic transportation technology of coarse particles.

**Keywords:** pipeline hydraulic transportation; coarse particles; lattice Boltzmann method; discrete element method; fluid-structure interaction





## 1. Introduction

Transportation is the process of moving people and goods from one location to another using various delivery methods. It is an essential component of modern society, and its efficiency is critical to the development of the economy and society. The modern transportation system encompasses roads, railways, waterways, airways, and pipelines, each playing a significant role in facilitating the movement of goods and people. Pipeline hydraulic transportation is a technique that employs fluid flow to transport solid materials through pipelines. Despite its relatively recent development compared to other transportation methods, it has grown rapidly and is now a crucial aspect of modern transportation systems. This method is widely used in a range of industries, including metallurgy, coal, chemical manufacturing, and water conservation, owing to its numerous advantages, such as high transport capacity, low cost, small area coverage, and minimal susceptibility to natural conditions.

Pipeline hydraulic transportation involves multiple disciplines, including fluid mechanics, solid mechanics, and solid-liquid two-phase flow. The technology is very comprehensive, but due to the complex internal conditions of the pipeline, there are still many problems that need to be solved, especially the mutual influence of fluid and solid materials.

During the transportation process, the fluid causes the solid to deform and move, while the solid also affects the flow velocity and direction of the fluid. The solid-liquid two-phase flow in the pipeline essentially belongs to the fluid-structure interaction problem. Given the importance and complexity of pipeline hydraulic transportation, related research is a hot issue.

Experimental research on pipeline hydraulic transportation has received widespread attention, and many scholars have investigated the influence of macro parameters of fluids and solids on pipeline transportation, resulting in significant research achievements. Vlasák et al. [1] studied the movement characteristics of a coarse particles-water mixture flow in pipes, and the change of frictional pressure drops was explored by changing the pipe inclination angle. Matoušek et al. [2] measured the concentration distribution of ballotini in pipe flow and found that flows tended to be partially stratified. Zouaoui et al. [3] conducted experiments on the influence of large particles' physical characteristics on horizontal pipes and observed the relationship between pressure gradient forces and mixture velocity. Alihosseini et al. [4] studied the motion of sediments on sewer beds, and the results showed that the sediment size had a greater influence on critical velocity than the bed roughness. Kaushal et al. [5] modified the Karabelas model for the prediction of particle concentration distribution in horizontal pipes based on experimental data. Osra et al. [6] measured the head loss of pipeline flow through experiments and described the general rule of hydraulic gradient varying with solid concentration and pipeline inclination. Vlasák et al. [7] measured the velocity distribution of liquid and transport particles. At the same time, the motion of the individual particle was described. Spelay et al. [8] studied the influence of pipe inclination on the deposition velocity of particles with different diameters through experiments. Some scholars have conducted experimental research on stabilization and rheological properties of high concentration ore slurry by adding surfactants, such as coal ash and bottom ash, and obtained some feasible conclusions for engineering applications [9–12].

Numerical simulations of pipeline hydraulic transportation have also produced significant results, especially using coupled computational fluid dynamics (CFD) and the discrete element method (DEM). Xiong et al. [13] and Chen et al. [14] established a coupled CFD-DEM method and explored the internal structure of the flow. Yang et al. [15] simulated the transport characteristics of slurry shield, and the impacts of velocity and concentration on transport capacity were investigated. Akbarzadeh et al. [16] simulated the migration process of particles in rectangular pipes, and discussed the influencing mechanisms of buoyancy force, adhesion, and drag force on the movement mechanism of particles. Guo et al. [17] compared three common CFD-DEM models for simulating soil surface erosion and gave some reasonable suggestions when choosing different particle-fluid systems. Zhou et al. [18] simulated the conveying of solid particles in vertical pipes, and the results showed that the solid volume fraction and axial velocity of liquids and solids all had the maximum value in the middle of the pipeline. Januário et al. [19] simulated the critical deposition velocity of various ores, and the results were compared with the experimental data from other studies. Akhshik et al. [20,21] studied the transportation mechanism of fluid and cuttings in well-drilling through the CFD-DEM method.

The CFD-DEM method is able to simulate the collision behavior of coarse particles in solid–liquid two-phase flow and has also been applied to the study of erosion damage of particles on the inner walls of pipelines. By using the Rosin-Rammler particle distribution model, Wee et al. [22] improved the erosion damage prediction method of pipelines and predicted the erosion damage of pipelines under different influencing factors. Zolfagharnasab et al. [23] used CFD to study the erosion mechanism on a square duct and found a new erosion pattern different from that of common pipes. Varga et al. [24] simulated the erosion of a pipeline and studied the influence of particle shape on particle-wall impact, concluding that particle shape had an important influence on the erosion pattern. By simulating the erosion failure of horizontal pipes, Cheng et al. [25] concluded that the erosion rate and erosion area were mainly affected by the flow velocity and the viscosity of

liquid. Chen et al. [26] predicted the maximum erosion rate and location in elbows with different angles. Tao et al. [27] discussed the influencing factors of pipe erosion resistance through numerical simulation, and the internal relation between some parameters and pipe wall failure was revealed.

The coupled method of the lattice Boltzmann method (LBM) and DEM is also widely used in the study of solid-liquid two-phase flow. Cui et al. [28] used a coupled LBM-DEM method to numerically simulate the soil fluidization problem induced by pipeline leakage. The results showed that the LBM-DEM method can capture the leakage-soil interaction at the particle scale, and the LBM-DEM method was considered to be a promising tool for simulating the internal fluidization phenomenon. Song et al. [29] simulated the motion of particle deposition and resuspension of a pre-deposited dynamic membrane and conducted a microscale quantitative analysis of the process. Yang et al. [30] presented a parametric study for immersed granular flows using the LBM-DEM method, and the results showed that LBM-DEM can effectively describe the dynamic characteristics of friction-dominated and densely packed immersed granular flows. Tran et al. [31] simulated the granular soil at the front end of the erosion pipe and explored the backpropagation mechanism of erosion. Zhou et al. [32] used the LBM-DEM method to simulate the initialization and diffusion process of erosion particles under hydraulic flow conditions and revealed the micromechanical process of internal erosion.

In this paper, the coupled method of LBM and DEM is used to simulate the process of pipeline hydraulic transportation, and the motion characteristics of coarse particles in pipelines under different conditions are studied. Compared with the CFD-DEM method, this method is closer to the physical reality in dealing with the fluid-structure interaction problem. The disadvantage of LBM is that it requires higher computational cost, but it has great potential for the study of large flow field problems due to its advantages, such as clear physical meaning, ability to handle complex boundary conditions, and suitability for parallel computing.

Pipeline hydraulic transportation is widely used in many industrial fields and is of great significance for promoting the development of related industries. However, the interaction between fluids and solids inside the pipeline, as well as the restriction of fluid and solid movement by the inner wall of the pipeline, make the solid-liquid two-phase flow in the pipeline highly complex, which seriously hinders the development of pipeline hydraulic transportation technology. To investigate the motion mechanism of coarse particles in the pipeline, a numerical simulation method based on LBM-DEM has been developed and verified. This method can simulate the fluid-structure interaction problem in the pipeline and can visually demonstrate the mutual influence between particles and fluid flow in the pipeline. This method has clear physical meaning and can reveal the inner mechanism of fluid flow and solid motion at a deeper level, which has significant advantages for the in-depth investigation of pipeline transportation problems, and it is important for promoting the technical development of pipeline hydraulic transportation.

This paper is structured as follows: the coupled LBM-DEM method is introduced in Section 2. Section 3 validates the proposed method by comparing the simulation results with experimental results of the pipeline hydraulic transportation test. The influence of different initial conditions on coarse particle pipeline movement is studied in Section 4. Conclusions are drawn in Section 5.

## 2. Methodology

A simulation method that models the hydraulic transportation of coarse particles through a pipeline has been developed using the Visual Studio platform and Fortran language. The simulation utilizes the lattice Boltzmann method to simulate fluid flow, the discrete element method to simulate particle movement, and the immersed moving boundary method (IMB) to simulate interactions between fluid and solid particles. The simulation data is visualized using the 2019 version of Tecplot software.

### 2.1. Lattice Boltzmann Method

The lattice Boltzmann method is a computational technique that represents fluid as a collection of discrete particles resting at regular lattices. These particles undergo collision and relaxation processes on lattice nodes following specific rules. The method accurately models the motion and evolution of fluid particles using the lattice Boltzmann equation, which is expressed as:

$$f_i(x + c_i \Delta t, t + \Delta t) - f_i(x, t) = \Omega_i(x, t), \tag{1}$$

in which, $i$ is the number of discrete velocities of the particle; $x$ is the position vector of the particle; $c_i$ is the discrete velocity vector of the particle; $\Delta t$ is the time step; $f_i(x, t)$ is the fluid distribution function; $\Omega_i(x, t)$ is the collision operator.

The DnQb model under the lattice Bhatnagar-Gross-Krook (LBGK) model is used, in which n denotes the spatial dimension, b denotes the number of velocity vectors, and the $\Omega_i$ is defined as:

$$\Omega_i = \frac{\Delta t}{\tau} \left[ f_i^{eq}(\rho, u) - f_i(x, t) \right], \tag{2}$$

in which, $\tau$ is the relaxation time; $\rho$ is the macroscopic fluid density; $u$ is the macroscopic fluid velocity; $f_i^{eq}(\rho, u)$ is the equilibrium distribution function, and it is defined by:

$$f_i^{eq}(\rho, u) = \omega_i \rho \left[ 1 + \frac{c_i \cdot u}{c_s^2} + \frac{(c_i \cdot u)^2}{2c_s^4} - \frac{u^2}{2c_s^2} \right], \tag{3}$$

in which, $\omega_i$ is the weighting factor; $c_s$ is the fluid speed of sound and $c_s = c/\sqrt{3}$; $c$ is the lattice speed.

In this paper, the D2Q9 model is used to study the two-dimensional problem, and the velocity is discretized to nine directions ($c_i$, $i$ = 0, 1, 2, 3, 4, 5, 6, 7, 8), as shown in Figure 1. The weighting factor $\omega_0 = 4/9$, $\omega_{1,2,3,4} = 1/9$, $\omega_{5,6,7,8} = 1/36$.

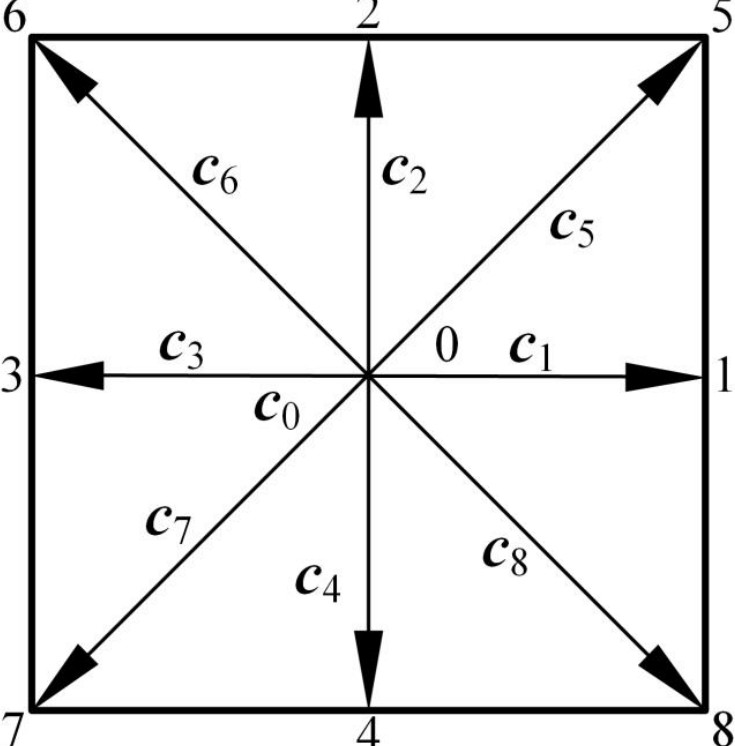

**Figure 1.** The D2Q9 model.

The column vectors of matrix $E$ represent the discrete velocity vector $c_i$, which is written as:

$$E = c \begin{bmatrix} 0 & 1 & 0 & -1 & 0 & 1 & -1 & -1 & 1 \\ 0 & 0 & 1 & 0 & -1 & 1 & 1 & -1 & -1 \end{bmatrix}, \tag{4}$$

After obtaining the particle distribution function, the macroscopic density and velocity at each point of the flow field can be calculated as:

$$\rho = \sum_{i=0}^{8} f_i(x, t), \tag{5}$$

$$u = \frac{1}{\rho} \sum_{i=0}^{8} c_i f_i(x, t). \tag{6}$$

### 2.2. Discrete Element Method

The discrete element method is commonly used to handle discontinuous problems and has wide applications in simulating discrete systems. In the DEM, each discrete particle is treated as a rigid body with mass, and its translation and rotation follow Newton's second law of motion. For each discrete element, the equilibrium equation can be expressed as:

$$M\ddot{a} + C\dot{a} + Ka = f, \tag{7}$$

in which, $a$ is the displacement; $\dot{a}$ is the first derivative of displacement with respect to time; $\ddot{a}$ is the second derivative of displacement with respect to time; $M$ is the mass matrix; $C$ is the damping matrix; $K$ is the stiffness matrix; $f$ is the applied force.

The interaction force between particles is generated through their contact. Therefore, being able to identify when particles are in contact is crucial for determining the degree of overlap between them. In this work, the no binary searching algorithm is used for particle contact detection, which has advantages such as high efficiency and small memory occupation compared with the particle contact detection algorithm based on the binary detection method [33].

### 2.3. Immersed Moving Boundary Method

The immersed moving boundary method is an LBM-DEMbased method for dealing with fluid-structure interaction problems [34]. It involves a well-defined classification of nodes into fluid, solid, and boundary nodes. Specifically, the solid nodes are further categorized into interior solid nodes and solid boundary nodes, while the fluid nodes are divided into pure fluid nodes and fluid boundary nodes. The local solid ratio $\varepsilon_s$ is a parameter that represents the proportion of nodes occupied by solid particles, with a value of 0 for pure fluid nodes, and 1 for interior solid nodes. The solid boundary nodes and fluid boundary nodes have values ranging between 0 and 1, as illustrated in Figure 2.

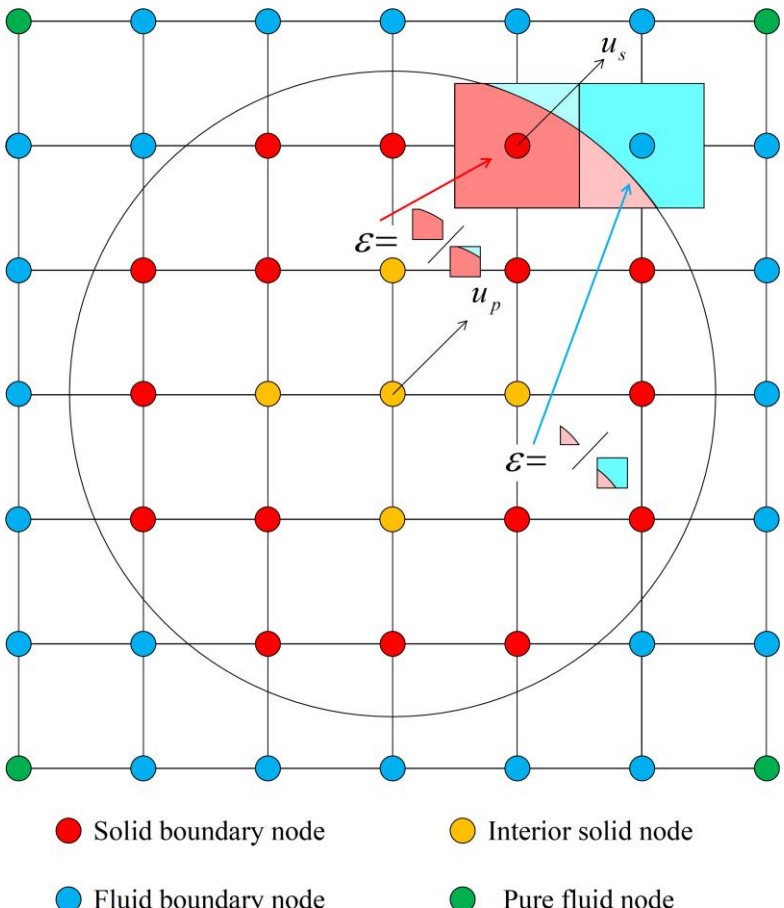

**Figure 2.** IMB nodes and the local solid ratio.

The IMB method extends the traditional lattice Boltzmann method by introducing the local solid ratio $\varepsilon_s$ and an additional collision term $\Omega_i^s$, allowing the LBM to smoothly transition between fluid dynamics at pure fluid nodes, and rigid body motion at interior solid nodes. When the external force term is neglected, the lattice Boltzmann equation modified by the IMB method is rearranged as:

$$f_i(\boldsymbol{x} + \boldsymbol{c}_i \Delta t, t + \Delta t) - f_i(\boldsymbol{x}, t) = \frac{\Delta t}{\tau}(1 - B)\left[f_i^{eq}(\rho, \boldsymbol{u}) - f_i(\boldsymbol{x}, t)\right] + B\Omega_i^s, \tag{8}$$

in which, $B$ is the weighting function depending on the local solid ratio $\varepsilon_s$; $\Omega_i^s$ is an additional collision term based on the bounce-back rule of the nonequilibrium part, and the expressions are given by:

$$B = \frac{\varepsilon_s(\tau - 0.5)}{(1 - \varepsilon_s) + (\tau - 0.5)}, \tag{9}$$

$$\Omega_i^s = f_{-i}(\boldsymbol{x}, t) - f_i(\boldsymbol{x}, t) + f_i^{eq}(\rho, \boldsymbol{u}_s) - f_{-i}^{eq}(\rho, \boldsymbol{u}), \tag{10}$$

in which, $f_{-i}$ is the bounce-back state of $f_i$; $\boldsymbol{u}_s$ is the velocity of the solid node, and the solid velocity at each node inside the solid is equal to the particle center velocity $\boldsymbol{u}_p$ without considering the rotation of the solid particle.

The effect of the fluid on the solid particle is represented by the lattice nodes covered by the solid particle, which all generate hydrodynamic forces and torques on the solid particle; the hydrodynamic force $\boldsymbol{F}$ and the torque $\boldsymbol{T}$ of the fluid on a solid particle are

the sum of the forces and torques generated by all the lattice nodes covered by the solid particle, which are determined as:

$$F = \sum_{j=1}^{n} \left( B_j \sum_{i=0}^{8} \Omega_i^s c_i \right),$$ (11)

$$T = \sum_{j=1}^{n} \left[ (x_j - x_p) B_j \sum_{i=0}^{8} \Omega_i^s c_i \right],$$ (12)

in which, $n$ is the number of the lattice nodes covered by the solid particle; $x_j$ is the position vector of the $j$th lattice node; $x_p$ is the position vector at the center of the solid particle.

### 2.4. Calculation Process

The calculation flow chart of the coupled LBM-DEM algorithm is shown in Figure 3. $\Delta t_L$ is the time step of LBM, and $\Delta t_D$ is the time step of DEM; $\Delta t_L$ is generally larger than $\Delta t_D$, so multiple DEM subcycles will be carried out in one LBM cycle.

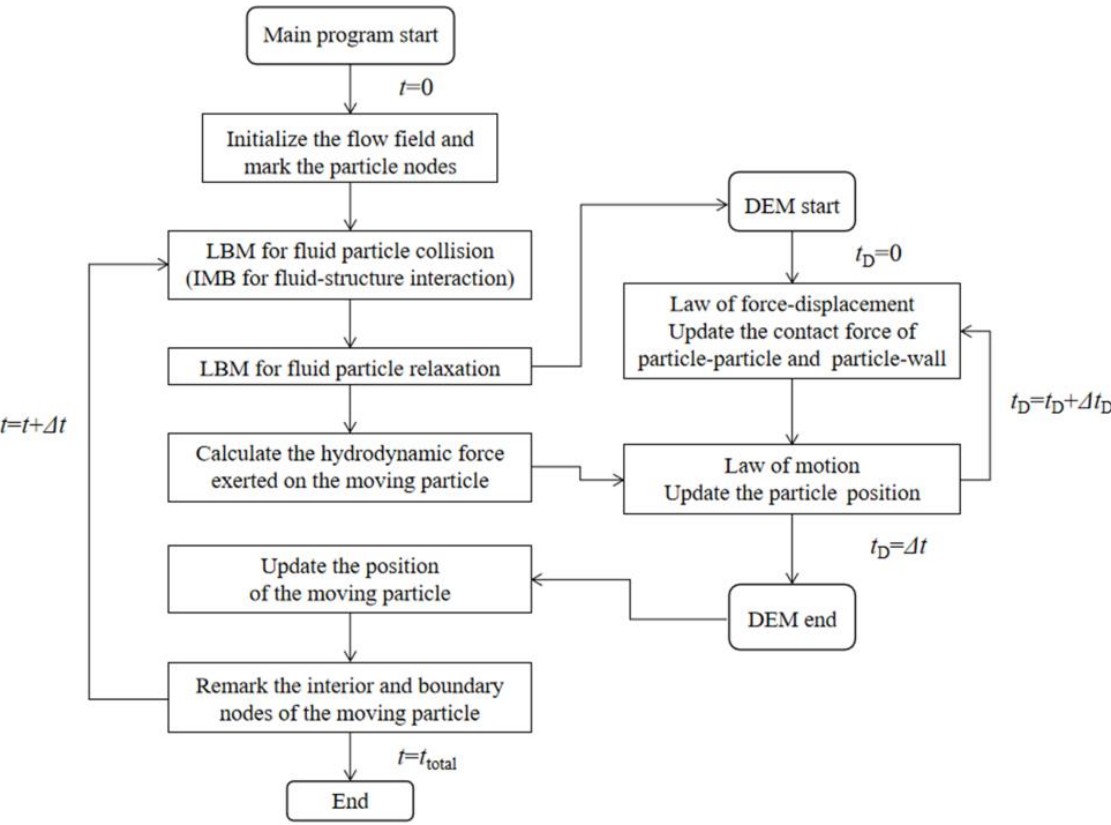

**Figure 3.** The flowchart of the LBM-DEM method.

### 3. Method Validation

#### 3.1. Pipeline Hydraulic Transportation Test

The experimental system consists of the stirring system, observation system, and water circulation system. The experimental setup is shown in Figure 4. The pipeline diameter is 0.1 m, and the water flow will enter the observation section at a steady pipeline flow velocity after passing through the stirring system. The flow rate in the pipeline is controlled by a frequency converter, whose range is 0~50 Hz and accuracy is 0.001 Hz. Meanwhile, the flow rate in the pipeline is measured by an electromagnetic flowmeter, whose range is 0~38 L/s and accuracy is 0.01 L/s. Solid particles are spherical grinding beads of 35% zirconia, and each particle has the same physical and mechanical parameters. Coarse particles are fed into the observation system from the feed box and pass through a

horizontal observation section with a length of 4 m. A pressure sensor is installed at the bottom of the pipeline every 1 m to record the pressure changes inside the pipeline, and the range of the pressure sensor is 0~50 kPa. A high-speed camera is set up between pressure measuring points 3 and 4 to capture the motion of coarse particles in the pipeline. The water flow can be recycled into the stirred tank through the water circulation system.

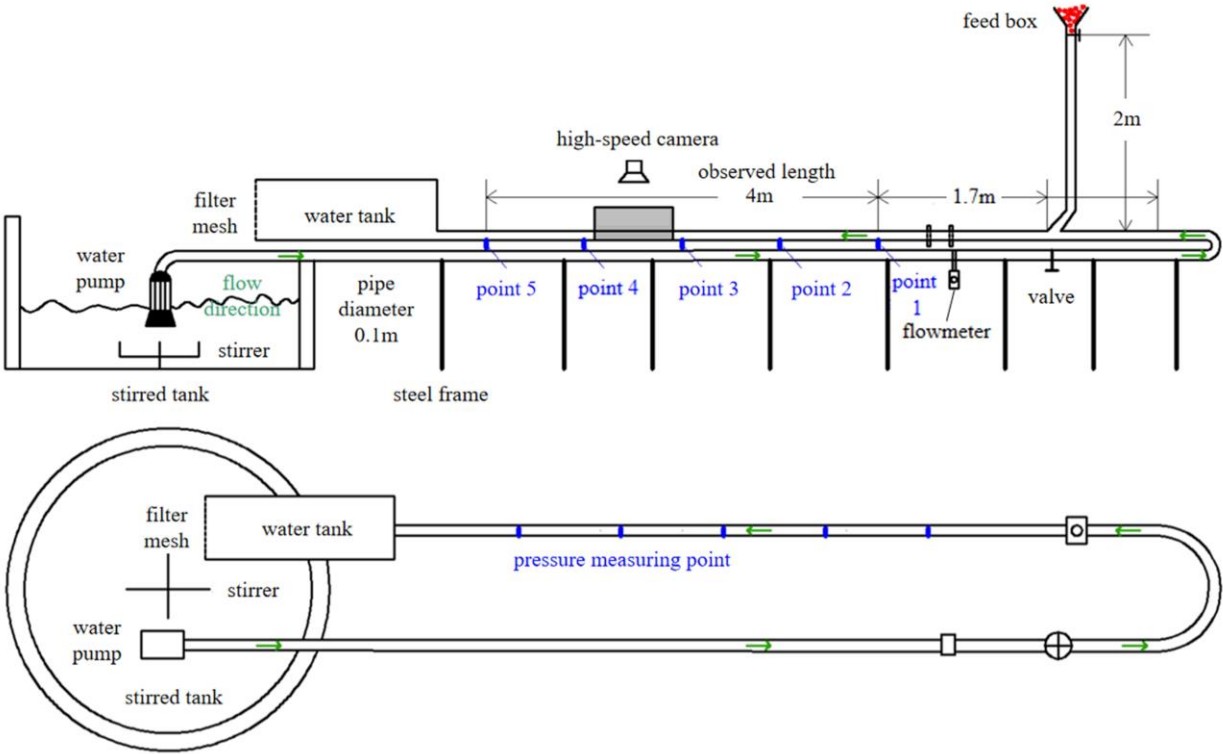

**Figure 4.** The schematic diagram of the experimental system.

The hydraulic transport process of coarse particles in a horizontal pipeline with different particle sizes and initial pipeline flow velocity is investigated using this test device. This paper focuses on the development of the simulation algorithm, and the experimental results are only for verification, so there will be no comprehensive discussion regarding experimental results.

### 3.2. Parameters of Numerical Simulation

The motion process of coarse particles in the pipeline is simulated using the established LBM-DEM method, and some of the simulation parameters are shown in Table 1.

**Table 1.** Simulation parameters.

| Parameter | Value | Parameter | Value |
|---|---|---|---|
| Particle density/$(kg/m^3)$ | 2650 | Fluid density/$(kg/m^3)$ | 1000 |
| Friction coefficient | 0.8 | Kinematic viscosity/$(m^2/s)$ | $1.0 \times 10^{-6}$ |
| Particle normal stiffness/$(N/m)$ | $2.0 \times 10^7$ | Particle tangential stiffness/$(N/m)$ | $2.0 \times 10^7$ |
| Coefficient of restitution | 0.6 | Damping coefficient | 0.51 |

In order to improve the efficiency of the calculation, it is necessary to convert the parameters related to time and space from physical coordinates to lattice coordinates; that is, dimensionless processing of these physical quantities. In this paper, the lattice spacing $dx$ is 1 mm, and the time step $\Delta t$ is $1.0 \times 10^{-5}$ s. The pipeline diameter is 0.1 m, which is converted into a lattice unit of 100. If the simulation time is 1.0 s, it transfers to a lattice unit of 100,000.

According to the difference of Reynolds numbers, the flow state of a fluid can be divided into laminar flow and turbulent flow. However, in the actual pipeline transportation, the flow state of fluid is mostly turbulent flow. The large-eddy simulation (LES) is a common method to simulate turbulence. Yu et al. [35] selected the one-parameter Smagorinsky subgrid model, in which the Reynolds stress tensor is only related to the local strain rate, and they introduced the turbulence model based on LES into the lattice Boltzmann equation. In this method, the lattice Boltzmann equation is expressed in filtered form as:

$$\widetilde{f}_i(\boldsymbol{x} + \boldsymbol{c}_i \Delta t, t + \Delta t) - \widetilde{f}_i(\boldsymbol{x}, t) = \frac{\Delta t}{\tau_*} \left[ \widetilde{f}_i^{eq}(\rho, \boldsymbol{u}) - \widetilde{f}_i(\boldsymbol{x}, t) \right], \tag{13}$$

in which, $\widetilde{f}_i$ is the fluid distribution function of the resolved scales; $\widetilde{f}_i^{eq}$ is the equilibrium distribution function of the resolved scales; $\tau_*$ is the total relaxation time. Feng et al. [36] demonstrated the feasibility of this extended LBM method for turbulence simulations.

The distribution of flow velocity in the pipeline has an important effect on the movement of coarse particles in the pipeline. Nikuradse [37] obtains a relatively simple exponential velocity distribution formula to describe the horizontal velocity distribution of turbulence in a smooth circular pipeline through experiments, and the empirical formula is defined as:

$$\frac{u_x}{u_{x\max}} = \left( \frac{y}{R} \right)^{1/n}, \tag{14}$$

in which, $u_x$ is the fluid velocity in the horizontal direction; $u_{x\max}$ is the maximum fluid velocity in the horizontal direction; $R$ is the radius of the pipeline. The value of $n$ depends on the Reynolds number, the larger the Reynolds number, the larger the value of $n$, and the flatter the velocity distribution curve. The value of $n$ is taken as 4.6 in this article.

Taking the initial horizontal velocity of pipeline flow as 1.0 m/s and the lattice spacing dx = 1 mm as an example, the initial horizontal velocity distribution of pipeline flow along the y direction is shown in Figure 5. In the horizontal pipe, the initial velocity of the flow at the upper and lower pipe walls is 0. With the approach to the central axis of the pipeline, the initial velocity increases gradually, and the velocity distribution curve becomes more and more gentle, until the velocity reaches the maximum at the central axis of the pipeline.

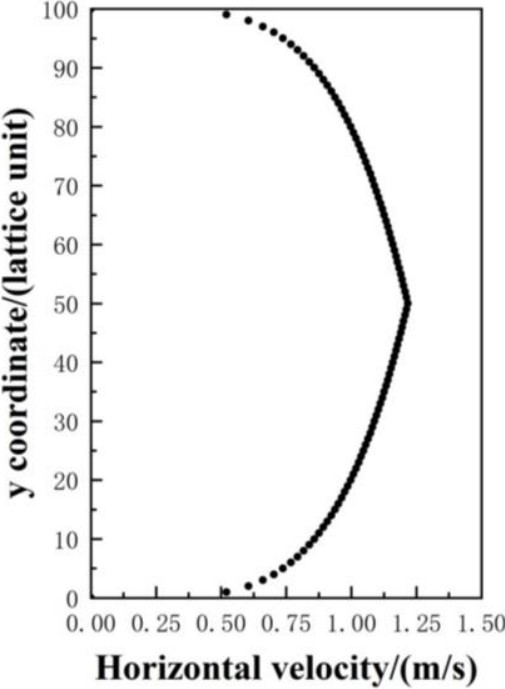

**Figure 5.** The initial horizontal velocity distribution of the pipeline flow.

### 3.3. Comparison of Experimental and Simulated Results

The movement process of 35% zirconia spherical grinding beads, with a particle diameter of 20 mm, at the initial pipeline flow velocity of 1.0 m/s is observed through the experimental system shown in Figure 4, and the horizontal velocities of 25 particles are recorded in the observation section. The experimental data are shown in Figure 6a, and the average velocity of each particle is shown in Figure 7. The same condition is also simulated using the established coupled LBM-DEM method, and the simulation results are shown in Figure 6b. Considering that 25 particles are uniformly put into the feed box, only 5 particles were arranged for simulation in order to save computing resources, and the initial interval between two adjacent particles is equal to the radius of a particle (r = 10 mm). The pipeline diameter of the simulated horizontal pipeline is 0.1 m and the pipeline length is 0.5 m. The inlet and outlet of the pipeline adopt the circulation boundary, and the pipeline flow and coarse particles flowing out from one side of the pipeline will re-enter from the other side of the pipeline. At the initial moment, the velocity distribution of pipeline flow is shown in Figure 5, and the velocity image and the particle distribution of the pipeline are shown in Figure 8.

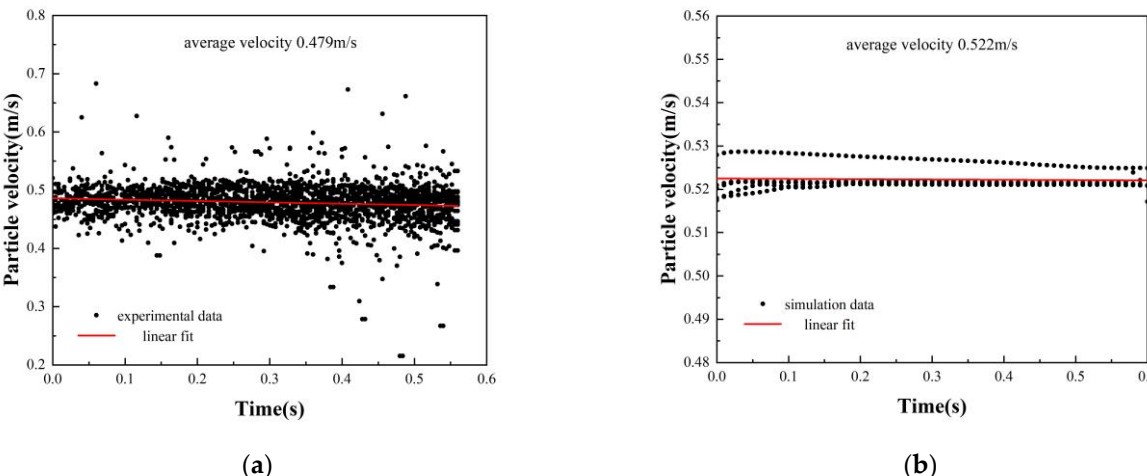

(**a**)                                                                                           (**b**)

**Figure 6.** Comparison of experimental and simulation results. (**a**) Experimental results. (**b**) Simulation results.

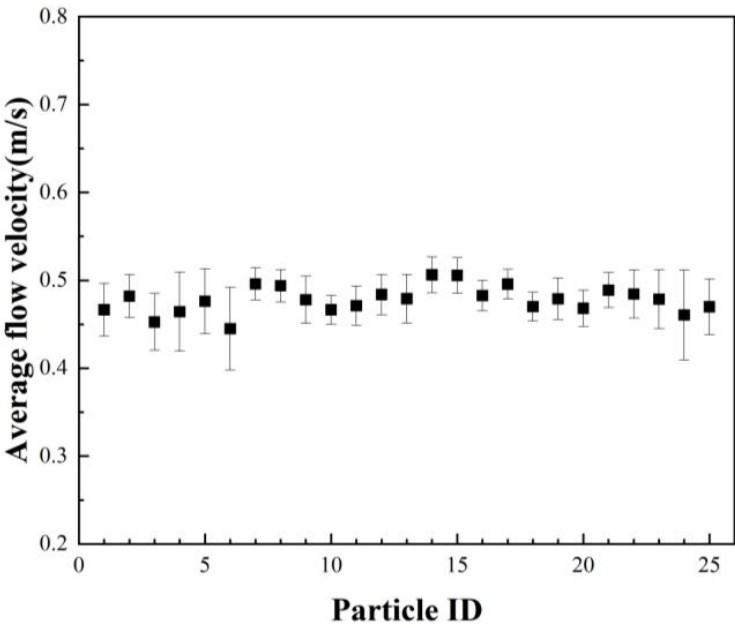

**Figure 7.** The average velocity of each particle in the experiment.

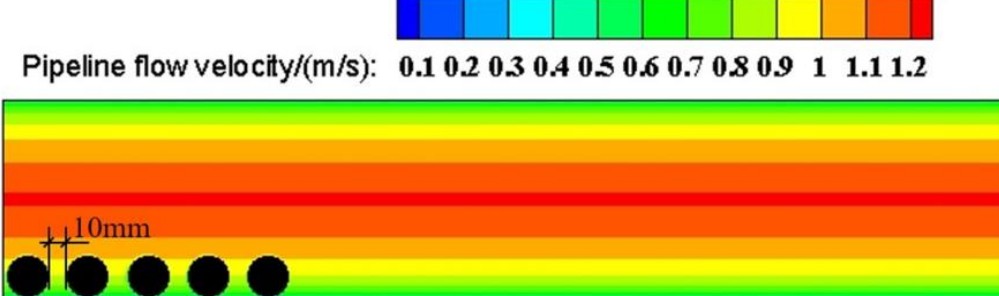

**Figure 8.** The initial state of the simulation.

As can be seen from Figure 6, the average horizontal velocity of coarse particles measured in the experiment is 0.479 m/s, and the average horizontal velocity of coarse particles simulated is 0.522 m/s, with an error of 8.98%. The simulation results are consistent with the experiment results, which verifies the feasibility of this LBM-DEM numerical method for simulating the pipeline transportation process of coarse particles.

## 4. Influence of Different Initial Conditions on Coarse Particle Pipeline Movement

Solid particles inside a pipeline are subject to the drag force exerted by the fluid. Once the drag force exceeds the threshold force required to initiate particle motion, the particles will start to move. Fei et al. [38] have classified particle motion into two categories, namely, bedload motion and suspension motion, based on the type of forces and energy consumed by the particles. In a horizontal pipeline, coarse particles undergo a transition from a static state to bedload motion and eventually to suspension motion as the intensity of water flow increases.

Bedload motion is the primary form of motion exhibited by coarse particles in a horizontal pipeline. In this type of motion, the particles move at a much slower average velocity than that of the fluid, and most of them move along the bottom of the pipe. The energy required for bedload motion is derived from the potential energy of the water flow. During bedload motion, some particles move forward in a jumping motion known as saltation. Suspension motion is another type of motion exhibited by coarse particles in a horizontal pipeline. In this form of motion, the particles are suspended in the water flow, and their average velocity is almost identical to that of the fluid. The energy required for suspension motion is obtained from the turbulent kinetic energy.

### 4.1. Influence of Pipeline Flow Velocity on Motion Form of Coarse Particles

To investigate the impact of pipe flow velocity on the motion of coarse particles, simulations are conducted to observe the motion of five 12 mm diameter coarse particles in a horizontal pipe under various initial flow velocities. At the initial moment, the particles are arranged uniformly along the bottom of the pipe, and labeled particle 1, 2, 3, 4, and 5 from left to right. The distance between any two adjacent particles is equal to the particle radius (r = 6 mm). The initial pipeline flow velocities are set at 1.0, 1.5, 2.0, 2.5, and 3.0 m/s, respectively. The motion of the particles under different flow velocities is presented in Figure 9. In order to identify the motion of the saltation particles, the first saltation particle is marked in red, the second in yellow, the third in green, and the nonsaltation particles remain in black.

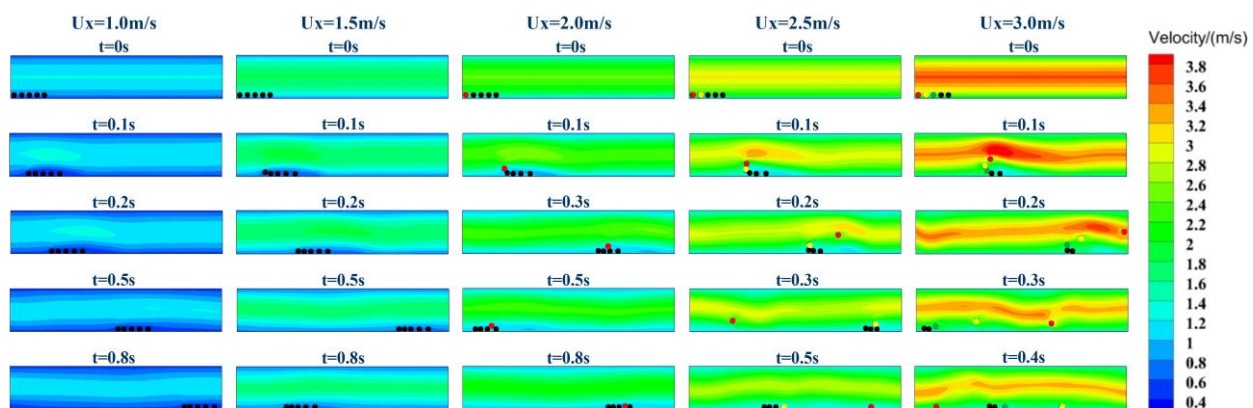

**Figure 9.** Movements of 12 mm diameter particles at different pipeline flow velocity.

As can be seen from Figure 9, at an initial flow velocity of 1.0 m/s, the particles exhibit bedload motion along the bottom of the pipe, and no saltation particle is observed. During the motion process, the particles on the left gradually converge and form a particle swarm. This phenomenon occurs due to the resistance of the left-side particles to the flow, causing the drag force on the right-side particles to be relatively small, and their velocity to be slightly lower than that of the left-side particles. As a result, the particles on the left converge into a particle swarm, increasing their resistance to the water flow until a steady state is reached. At an initial flow velocity of 1.5 m/s, the leftmost particle (particle 1) jumps upward briefly but falls back quickly without crossing the front particles. Ultimately, all five particles gather at the bottom of the pipe, exhibiting bedload motion.

At an initial flow velocity of 2.0 m/s, the force of the water on particle 1 is strong enough to lift it off the bottom of the pipe. As a result, particle 1 moves above particle 2 and starts rolling forward along the upper surface of the particle swarm formed by particles 2, 3, and 4. Eventually, particle 1 falls between particles 4 and 5, and the five particles gather closely together to form a particle swarm. This swarm continues to move along the bottom of the pipe as bedload.

When the initial flow velocity increases to 2.5 m/s, both particles 1 and 2 are lifted off the bottom of the pipe. Particle 1 jumps towards the central axis of the pipe, where the flow velocity is the highest, and is surrounded by the water there. As a result, its horizontal velocity is much higher than that of other particles at the bottom. Eventually, particle 1 falls back to the bottom of the pipe for bedload motion, landing far ahead of the other particles. Meanwhile, particle 2 jumps and rolls towards particle 3, eventually falling ahead of particle 5 and forming a particle swarm with particles 3 and 4 for continued bedload motion.

When the initial flow velocity continues increasing to 3.0 m/s, particles 1, 2, and 3 all saltate. Particles 1 and 2 saltate to the location with higher flow velocity and finally fall to the bottom of the pipe and make the bedload motion as a single particle, while particle 3 jumps above particles 4 and 5, and finally falls in front of particle 5 and forms a particle group with particles 4 and 5 to make the bedload motion.

The above analysis demonstrates that as the initial flow velocity increases, the force of the fluid on the particles increases as well. This results in more particles being lifted off the bottom of the pipe and reaching higher jump heights and distances. Additionally, as shown in Figure 9, the saltation particles cause the velocity contour of the pipe flow to bulge upward, leading to a sharp decrease in pipe flow energy and velocity. However, when the initial flow velocity is low and no saltation occurs, the energy consumption of the pipe flow is lower, resulting in a slower decrease in flow velocity.

When the fluid flows over the surface of a static particle in the pipe, the velocity at the top of the particle is higher than that at the bottom. According to Bernoulli's theorem, the flow pressure at the top of the particle is lower than at the bottom, creating a pressure difference that generates a vertical uplift force on the particle. Additionally, the particle

experiences its own gravity as well as the buoyancy of the water, which cancel out to create the effective gravity of the particle. If the uplift force exceeds the effective gravity, the particle will move upward, and saltation occurs. Conversely, if the uplift force is less than the effective gravity, the particle will move downward and eventually settle back on the bottom of the pipe. As the initial flow velocity of the pipe increases, the difference in velocity between the upper and lower surfaces of the particle also increases, resulting in a greater uplift force on the particle. Therefore, more saltation particles will appear. This conclusion is consistent with the research results of Vlasák et al. [1,7].

*4.2. Influence of Particle Diameter on Motion Form of Coarse Particles*

To investigate the impact of particle diameter on the motion of coarse particles, simulations were conducted at an initial flow velocity of 3.0 m/s using particles of different diameters. Five coarse particles were labeled as particle 1, 2, 3, 4, and 5 from left to right, with particle diameters of 10, 12, 14, 16, 18, and 20 mm, respectively. The simulation results are shown in Figure 10. To distinguish the motion of saltation particles, the first, second, and third saltation particles are marked in red, yellow, and green, respectively, while the nonsaltation particles remain black.

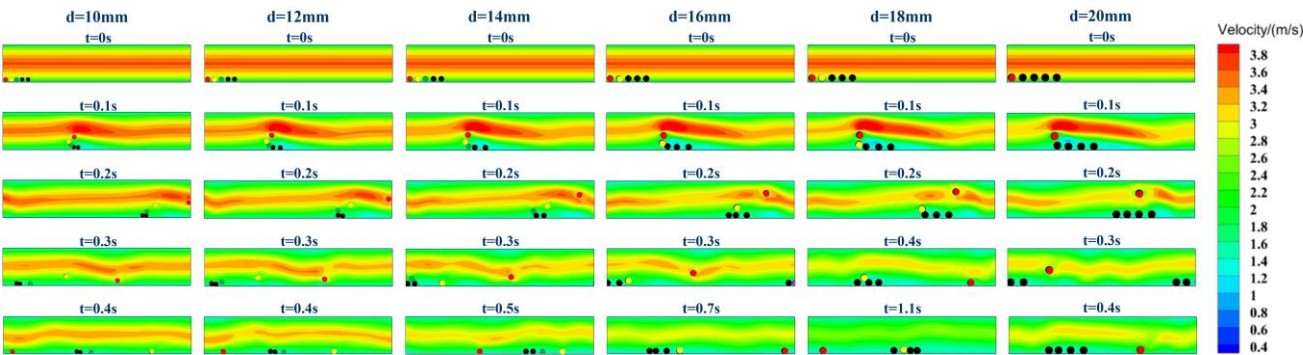

**Figure 10.** Movements of particles with different diameters at the flow velocity of 3 m/s.

Figure 10 shows that particles 1, 2, and 3 all undergo saltation when the particle diameters are 10, 12, and 14 mm, whereas only particles 1 and 2 undergo saltation when the particle diameters increase to 16, 18, and 20 mm. The simulation results indicate that larger particles are less likely to undergo saltation at the same initial flow velocity, as they require a greater uplift force and higher pipeline flow velocity to initiate saltation. Furthermore, even when particles of different diameters undergo saltation, those with larger diameters tend to exhibit fewer saltation events, lower jump heights, and shorter jump distances compared to those with smaller diameters.

In order to more intuitively show the influence of particle diameter on the motion form of coarse particle, the number of saltation particles at different pipeline initial flow velocities and particle diameters is counted, as shown in Table 2. The data in the table show the characteristics of the row-echelon form matrix; that is, when the initial flow velocity of the pipeline is the same, the number of saltation particles with larger diameters is always less than or equal to that of particles with smaller diameters.

**Table 2.** Number of saltation particles.

|  | 1.0 m/s | 1.5 m/s | 2.0 m/s | 2.5 m/s | 3.0 m/s |
|---|---|---|---|---|---|
| 10 mm | 0 | 1 | 1 | 2 | 3 |
| 12 mm | 0 | 1 | 1 | 2 | 3 |
| 14 mm | 0 | 0 | 1 | 2 | 3 |
| 16 mm | 0 | 0 | 1 | 1 | 2 |
| 18 mm | 0 | 0 | 1 | 1 | 2 |
| 20 mm | 0 | 0 | 0 | 1 | 1 |

To better understand the effect of particle diameter on saltation, simulations are conducted to determine the minimum pipeline flow velocity required for particle 1, with different diameters, to saltate. The results are shown in Figure 11, which clearly shows that the required pipeline flow velocity for saltation increases with particle diameter. The relationship between the two variables is highly linear, indicating that particle diameter is a critical factor in determining the critical velocity for saltation.

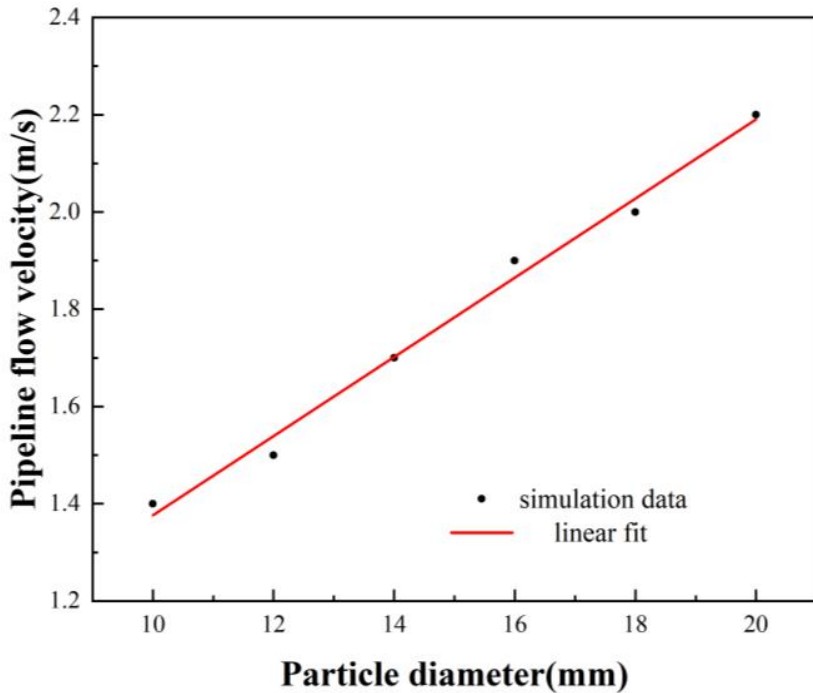

**Figure 11.** Critical pipeline flow velocity of saltation particles with different diameters.

### 4.3. Influence of Pipeline Flow Velocity and Particle Diameter on Coarse Particle Velocity

Under the influence of pipe flow, coarse particles initially in a static state will eventually reach a stable state when the velocity of each particle becomes constant. Saltation particles, on the other hand, reach the higher velocity area of the pipe flow, and their horizontal velocity is much higher compared to the particles moving in the low velocity area at the bottom of the pipe. Consequently, the time required for saltation particles to reach the steady state is much longer than the particles at the bottom of the pipe. In order to resist the impact of the water flow, the particles at the bottom of the pipe gradually come into contact to form a particle swarm, and the velocity of each particle in the particle swarm is almost the same. Hence, it can be treated as a single entity.

To investigate the velocity change of coarse particles from static to a stable state, the simulation results of pipe flow velocities of 1.0 m/s and 1.5 m/s are analyzed. At these two flow velocities, no saltation particles appear or particles saltate to a lower height, resulting in a small velocity difference between particles at the same moment. Therefore, these particles could be studied as a whole. In contrast, particles that saltate to higher heights differ significantly from those at the bottom of the pipe, and thus cannot be studied as a unified whole.

The motion process of coarse particles from static to stable state is simulated with different diameters of coarse particles at the initial flow velocity of 1.0 m/s and 1.5 m/s. Coarse particle diameters are set as 10, 12, 14, 16, 18, and 20 mm, and the velocity variation curves of coarse particles from static to stable state at pipe flow velocities of 1.0 m/s and 1.5 m/s are shown in Figure 12a,b, respectively. From Figure 12a, it can be observed that when the particle diameter is 10 mm and 12 mm, the particle velocity gradually increases and the velocity curve has a "C" shape, which is consistent with the characteristics of a concave function. As the particle diameter increases to 14 mm, 16 mm, 18 mm, and 20 mm,

the velocity curve changes significantly, and the shape of the curve gradually changes from "C" to "S" shape, indicating a change from a concave to a convex function. Therefore, it can be concluded that the particle diameter has a significant impact on the velocity change of coarse particles from static to stable state.

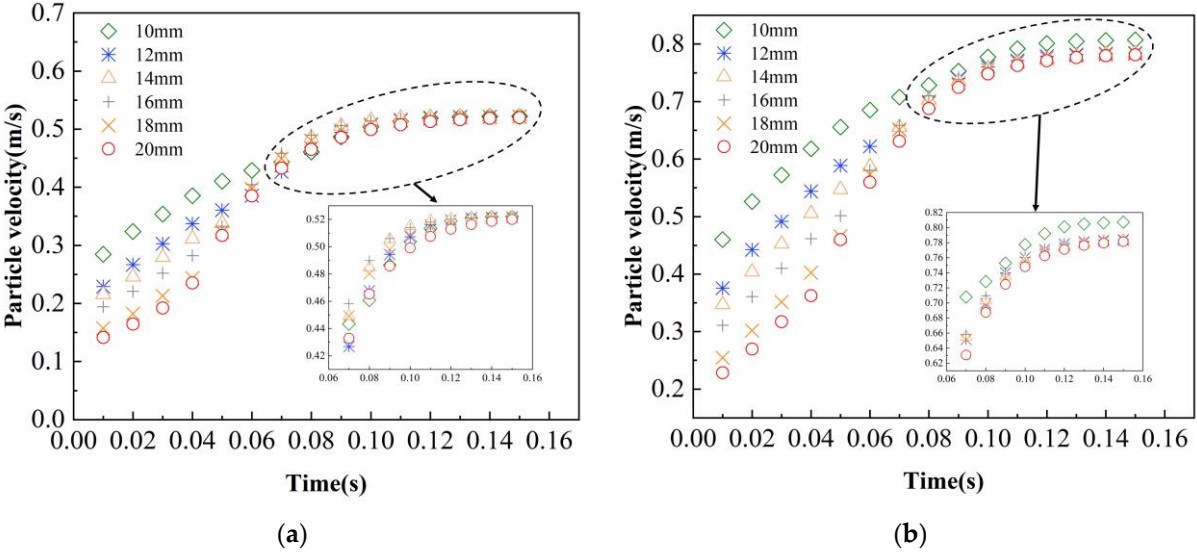

**Figure 12.** Change of the particle velocity with different particle diameters. (**a**) Pipeline flow velocity 1.0 m/s. (**b**) Pipeline flow velocity 1.5 m/s.

In Figure 12b, compared with Figure 12a, as the initial flow velocity of the pipeline increases from 1.0 m/s to 1.5 m/s, the particle velocity at the beginning and the particle stability velocity also increases, but the influence of particle diameter on particle velocity curve does not change.

In order to further investigate the influencing factors of particle stability velocity, the motion of coarse particles of 10 mm diameter in the pipe is simulated with different initial velocities of the pipe flow. The initial velocity of the pipe flow is set to 1.0, 1.5, 2.0, 2.5, and 3.0 m/s. As shown in Figure 13, with the increase of the initial velocity of the pipe flow, the stability velocity of the coarse particles in the pipe also increases, and there is a significant linear relationship between them.

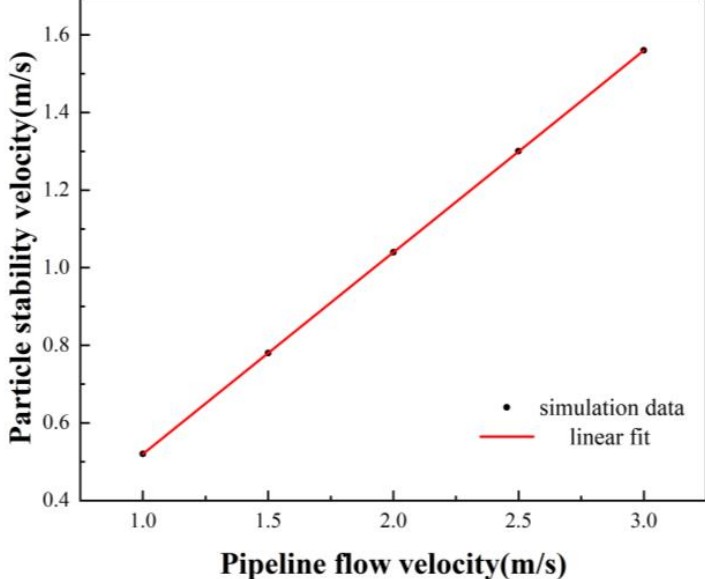

**Figure 13.** Particle stability velocity at different pipeline flow velocity.

## 5. Conclusions

In this study, the LBM-DEM simulation method is developed to investigate the hydraulic transport process of coarse particles in a pipeline. The simulation results are compared with experimental data to validate the method. The method is then applied to investigate the movement of coarse particles under various initial conditions. Based on the simulation results, several conclusions are drawn. (1) When the initial flow velocity of a pipeline is low, particles tend to move along the bottom and accumulate. As the initial flow velocity increases, some particles begin to saltate, causing a rapid drop in the pipe flow velocity. (2) The number and jump height or distance of saltation particles decrease as particle diameter increases. There is a linear correlation between particle diameter and the critical flow velocity required for saltation. (3) Particle diameter has a significant impact on particle velocity changes. As particles transition from a static to stable state, the particle velocity curve gradually changes from a concave "C" shape to a convex "S" shape, with this change becoming more pronounced as particle diameter increases. (4) The stability velocity of particles increases linearly with the initial velocity of the pipe flow, and the particle diameter has no significant effect on this relationship.

These findings indicate that manipulation of pipeline flow velocity and particle diameter can effectively regulate the movement of coarse particles in pipelines. In practical hydraulic transportation applications, appropriate pipeline flow velocity and particle diameter should be chosen based on specific circumstances. However, it is worth noting that this study is limited by the small number of particles used in simulations and the initial spacing between particles, which may have resulted in limited particle interactions. Future studies should explore the motion characteristics of closely-spaced particle clusters in pipelines.

**Author Contributions:** Conceptualization, T.Z. and Z.L.; formal analysis, Y.W. and W.N.; funding acquisition, T.Z. and Z.L.; methodology, Y.W. and Z.L.; software, Y.W. and T.Z.; validation, Y.W. and W.N.; investigation, Y.W., W.N. and T.Z.; resources, W.N., T.Z. and Z.L.; data curation, Y.W.; writing—original draft preparation, Y.W. and T.Z.; writing—review and editing, W.N., T.Z. and Z.L.; visualization, Y.W. and W.N.; supervision, T.Z. and Z.L.; project administration, W.N., T.Z. and Z.L. All authors have read and agreed to the published version of the manuscript.

**Funding:** This research is funded by the National Natural Science Foundation of China, grant number 12102294, the Natural Science Foundation of Shanxi Province, grant number 2021032124158, and the Research Project Supported by Shanxi Scholarship Council of China, grant number 2022-067.

**Data Availability Statement:** The datasets generated and analyzed in the current study are available by the authors on reasonable request.

**Conflicts of Interest:** The authors declare no conflict of interest.

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
