# Peer review of "Study on Characteristics of Pipeline Hydraulic Transportation of Coarse Particles Based on LBM-DEM Method"

_water, doi:10.3390/w15091717_

Round 1
Reviewer 1 Report
In this paper, a LBM-DEM based simulation method is used to study the movement process of coarse particles in the pipeline under different initial conditions. As far as I can see, this paper still needs a major revision before publication, the opinions are listed below:
1. In introduction, the application of LBM-EM method is not involved in the literature review.
2. The necessity and significance of this study is not well addressed.
3. The transfer between the lattice units and actual physical units is missing.
4. In line 215, it is said that the flow state is turbulent flow. Is the simulation carried out based on turbulent flow? The basic assumption of the simulation is in lack of explanation.
5. Fig. 11 is fuzzy and indistinct, and it needs to be redrawn.
6. Maybe the author should adopt different colors to indicating different particles, it is hard to distinguish the saltation of the specific particle.
7. The phenomenon of the particle saltation is described, but the underlying reason is not well explained.
8. The influence of the coarse particle numbers is not studied. Is the conclusion still valid for more particles? Or else the effective range of conclusion should be pointed out.
Author Response
English of the manuscript has been carefully checked by a native English-speaking colleague. Please see the attachment for the response.

Reviewer 2 Report
In present study, the coupled method of lattice Boltzmann method (LBM) and discrete element method (DEM) was used to simulate the process of pipeline hydraulic transportation, and the motion characteristics of coarse particles in pipeline under different conditions are studied. Compared with the CFD-DEM method, this method is closer to the physical reality in dealing with the fluid-structure interaction problem. The disadvantage of LBM is that it requires higher computational cost, but it has great potential for the study of large flow field problems due to its advantages such as clear physical meaning, ability to handle complex boundary conditions, and suitability for parallel computing. The descriptions of the methodology and simulation procedures have shown the feature to the research area. Generally the article, it explains what has been done and what has been discovered. The subject matter is interesting because the paper gives access to the information of this particular research. In summary, this is a piece of acceptable work but it needs some revisions. The paper can be accepted for publication, provided that the following points can be clarified.
1. The abstract should reflect the purpose of the study, the main results, the advantages of the approach used. In addition, nothing is said about the experiment, although much attention is paid to the description of the experiment in the manuscript.
2. Details of test rig components - technical features, measuring ranges and accuracy of the measuring devices of these instruments in Fig.4 should be given.
3. The conclusions can be further elaborated: what was the limitation of the study?, what do the study results imply for the larger stakeholders? (policy makers, practitioners, the economic sector, etc. address them one by one), what future study avenues are opened / suggested after this study?
4. Please provide the statistical test results. Please indicate how many rounds of trials were conducted? Please also report their variances. Please validate the proposed method on multiple datasets. Please add standard deviations to graphs.
5. The decision tree has the following assessment criteria, prediction accuracy, over-learning, complexity, processing scale and robustness. The authors should discuss based on these indicators.
6. The results were not at all compared with the research of other authors. It is difficult to evaluate the correctness of the experiments and the results without comparison. It is necessary to find as close as possible research oriented in terms of materials and parameters, because in the present form it is only the presentation of the results.
Round 2
Reviewer 1 Report
My comments have been responded in detailed, this manuscript can be recommended to publish in Water.